# No Effect of Lifestyle Intervention during Third Trimester on Brain Programming in Fetuses of Mothers with Gestational Diabetes

**DOI:** 10.3390/nu13020556

**Published:** 2021-02-08

**Authors:** Franziska Schleger, Katarzyna Linder, Louise Fritsche, Jan Pauluschke-Fröhlich, Martin Heni, Magdalene Weiss, Hans-Ulrich Häring, Hubert Preissl, Andreas Fritsche

**Affiliations:** 1Helmholtz Center Munich at the University of Tübingen/fMEG Center, Institute for Diabetes Research and Metabolic Diseases, 72076 Tübingen, Germany; linder.kasia@gmail.com (K.L.); louise.fritsche@med.uni-tuebingen.de (L.F.); martin.heni@med.uni-tuebingen.de (M.H.); hu.haereing@gmail.com (H.-U.H.); hubert.preissl@helmholtz-muenchen.de (H.P.); andreas.fritsche@uni-tuebingen.de (A.F.); 2German Center for Diabetes Research (DZD), 72076 Tübingen, Germany; 3Department of Internal Medicine IV, Division of Endocrinology, Diabetology and Nephrology, University Hospital Tübingen, 72076 Tübingen, Germany; 4Department of Obstetrics and Gynecology, University Hospital Tübingen, 72076 Tübingen, Germany; Jan.Pauluschke-Froehlich@med.uni-tuebingen.de (J.P.-F.); Magdalene.weiss@med.uni-tuebingen.de (M.W.); 5Department for Diagnostic Laboratory Medicine, Institute for Clinical Chemistry and Pathobiochemistry, University Hospital Tübingen, 72076 Tübingen, Germany; 6Department of Pharmacy and Biochemistry, Faculty of Science, University of Tübingen, 72076 Tübingen, Germany; 7Helmholtz Diabetes Center, Helmholtz Center Munich, German Research Center for Environmental Health (GmbH), Institute for Diabetes and Obesity, 85764 Neuherberg, Germany

**Keywords:** fetal MEG, oGTT, type 2 diabetes, maternal metabolism, fetal programming

## Abstract

Maternal metabolism and intrauterine conditions influence development of health and disease in offspring, leading to metabolic, physiologic, and/or epigenetic adaptation of the fetus. Maternal gestational diabetes (GDM) leads to higher incidence of obesity and type 2 diabetes in offspring. We have previously shown that fetuses of insulin-resistant mothers with GDM have a delayed reaction to auditory stimuli in the postprandial state, indicating a fetal central insulin resistance. We tested whether this effect could be influenced by a lifestyle intervention in mothers with GDM, including diet counselling and regular blood glucose measurements. We measured fetal brain activity over the course of a maternal glucose challenge, at two measurement time points (baseline at an average of 29 weeks of gestation and follow-up after 4 weeks) in mothers with GDM and mothers with normal glucose tolerance (NGT). Data from eight mothers were able to be included. Fetuses of GDM mothers showed longer latencies than those of NGT mothers postprandially at both measurement time points during the third trimester and did not show a difference in response patterns between baseline and after 4 weeks. Maternal postprandial blood glucose and insulin values did not change from baseline to follow-up either. While the overall intervention seems to have been effective, it does not appear to have influenced the fetal postprandial brain responses. This might have been because interventions for GDM take place relatively late in pregnancy. Future research should focus on maternal lifestyle interventions as early as possible during gestation, or even prenatally.

## 1. Introduction

Gestational diabetes (GDM) is associated with several short- and long-term consequences for both mother and child. Every fourth woman diagnosed with GDM is expected to have type 2 diabetes in 15 years [1,2]. Additionally, GDM is affecting the next generation by increasing the risk for perinatal complications [3] and the risk of becoming obese and developing diabetes later in life [4]. The pathogenesis of GDM is still not fully understood. It evolves when the pancreatic ß cells can not compensate increasing insulin resistance during pregnancy. Genetic and epigenetic factors have been shown to have an influence as well [5]. Focusing on possible interventions during pregnancy or prevention pre-pregnancy should be an important part of current research [6].

Treatment according to guidelines has been shown to improve the acute obstetric complications by reducing the risk of macrosomia [7,8], but the treatment impact of GDM on long-term consequences for mother and offspring is still not researched sufficiently.

Previously, we have shown that maternal insulin sensitivity moderates the latency of fetal brain responses to auditory stimulation during an oral glucose tolerance test (oGTT) [9]. Fetuses of mothers with GDM show slower postprandial event-related brain responses than those of metabolically healthy mothers, indicating an influence of GDM on fetal brain development [10].

In the present study, we investigated the influence of GDM treatment according to the therapy consensus guidelines of the German Diabetes Association (DDG) [11] on maternal metabolism and fetal brain activity during an oGTT.

Pregnant women with GDM were measured once during a standard diagnostic 75 g oGTT (between 27 and 31 weeks of gestation) and again after 4 weeks of treatment. Results were compared with a group of metabolically healthy pregnant women (with normal glucose tolerance, NGT). We aimed to investigate whether the difference in fetal brain activity between GDM and NGT subjects described in the previous study [10] was still present after a month of standard treatment including diet counselling and regular blood glucose measurements. We hypothesized that in the GDM group, maternal postprandial insulin sensitivity would improve during treatment that normalized glucose metabolism of the mother, and that fetal postprandial brain responses in GDM mothers would be faster, approaching those of the NGT group.

## 2. Materials and Methods

Fourteen healthy pregnant women with uncomplicated pregnancies, all part of the Tübingen PREG study cohort (NCT04270578), participated in this study. Inclusion criteria were singleton pregnancies, with reliably determined gestational age, 27+ weeks of gestational age (GA), maternal body mass index (BMI) below 40 kg/m^2^, and informed consent. GA at initial measurement ranged from 27 to 31 weeks (mean: 28.9 ± 1.3 weeks). Of the 14 participants, 6 had to be excluded from the final analysis (see Figure 1)—data of 5 could not be included because no fetal brain activity could be detected at the relevant postprandial measurement time point due to low signal-to-noise ratio of the relevant measurement. In one case, the intervention was not conducted according to study protocol because of non-compliance. Therefore, data from 8 women—4 with GDM and 4 with NGT—were analyzed.

Participants gave written informed consent before any measurements were made.

The Ethical Committee of the Medical Faculty of the University of Tübingen approved the study plan (project number 112/2016BO2).

The study course is displayed in Figure 1. There were two measurement time points: baseline and follow-up after 4 weeks. At baseline, a 75 g oGTT was conducted, and glucose and insulin levels were measured fasting and after 1 h and 2 h. Venous blood collections were preceded by fetal magnetoencephalographic (fMEG) recordings with fetal auditory stimulation (for more detail, see below) as an indicator for fetal brain insulin sensitivity.

Gestational diabetes was diagnosed on the basis of the 75 g oGTT according to criteria of the DDG [11]. In cases where GDM was diagnosed, women immediately received intervention.

This included structured consultation with a diabetologist, introduction to blood glucose self-monitoring (with instruction to measure and document fasting and postprandial blood glucose levels three times a day), as well as nutrition counselling by an experienced dietitian (with instruction to keep a food diary) and education about physical activity.

The intervention was targeted to achieve blood glucose levels at fast <5.56 mmol/L and 1 h postprandial <7.78 mmol/L. All patients were instructed to maintain a diet consisting of 40–50% preferably complex carbohydrates, 20% protein, and 30–35% fat, which was based on current German recommendations [12], One week after the initial consultation, the patient’s food diary and blood glucose measurements were evaluated and the decision for the subsequent treatment was made by the physician. Most of the blood glucose measurements (>50%) were within the therapeutic target, and therefore in none of the patients was insulin therapy required. The food diaries and results from frequent glucose measurements were controlled and discussed regularly, taking individual dietary preferences, daily routines, and socio-economic/cultural preconditions into consideration. Current physical activity levels were measured once with a short questionnaire [13], the benefits of physical activity were explained in detail, and the patient was strongly advised to follow an active lifestyle, but this was not controlled for.

Women with normal glucose tolerance were included as a control group and did not receive any intervention.

Four weeks after the baseline measurement, a second 75 g oGTT was conducted, with the same protocol. However, we cancelled the final fMEG measurement before the last blood draw as not to overstrain study participants at this later gestational age. Therefore, we included only the first two fMEG measurements (fasting and 1 h) for both time points (baseline and follow-up) in our analysis.

The following laboratory measurements were performed: Plasma glucose was measured with ADVIA 1800 autoanalyzer (Siemens Healthcare Diagnostics) by hexokinase method. Plasma insulin was analyzed using the ADVIA Centaur XP immunoassay system (Siemens AG). Plasma nonesterified fatty acid concentrations were measured enzymatically (WAKO Chemicals) using the ADVIA 1800 analyzer.

fMEG data were recorded with a dedicated 156-channel system (VSM MedTech Ltd., Port Coquitlam, BC, Canada). Fetal brain activity during auditory stimulation was recorded, with a measurement duration of 6 min. We presented 500 Hz sine tones in a standard oddball paradigm, interspersed with 750 Hz tones to prevent habituation [9,14]. Fetal auditory event-related responses (fAER) were analyzed (see [9] for more detail).

Statistical analyses were performed with SPSS (IBM SPSS Statistics for Windows, version 20.0) and with the software package JMP 13 (SAS Institute). Results with *p* < 0.05 were regarded as statistically significant. Repeated measures ANOVAs of oGTT (fasting, 1 h, 2 h) and group (GDM vs. NGT) on metabolic parameters were performed. Mann–Whitney tests were used for group comparisons and Wilcoxon tests for paired comparisons. Missing values were not replaced, with there being 2 missing values for habitual physical activity (HPA) Index (1 NGT; 1 GDM), 2 missing values for maternal blood glucose at 2 h at the follow-up time point (both NGT) and 1 missing value for fasting latency at baseline (in GDM). Power and sample size were calculated with PS Power and Sample Size Calculations, Version 3.0, January 2009.

## 3. Results

### 3.1. Participants

The anthropometric characteristics of four pregnant women with NGT and four pregnant women with GDM are shown in Table 1, and the metabolic characteristics are shown in Table 2. The groups did not differ significantly in maternal age, GA, parity, and neonatal birth weight. All subjects were Caucasian and had a high socioeconomic status (with seven subjects holding a college degree and one subject having completed a vocational training).

Women with GDM measured and documented their fasting blood glucose levels and postprandial blood glucose levels three times a day. During the course of the 4 weeks of lifestyle intervention, almost all documented blood glucose measurements were in the normal range (below 7.8 mmol/L). The mean fasting glucose was 5.0 mmol/L, and mean postprandial blood glucose was 6.6 mmol/L.

### 3.2. Maternal Blood Glucose and Insulin Levels

For maternal blood glucose and insulin values over the course of the oGTT (fasting, 1 h, 2 h) in both groups, see Table 2 and Figure 2. At the baseline time point, maternal blood glucose levels were significantly higher in the GDM group than in the NGT group—a repeated measures ANOVA showed a significant interaction of the group with oGTT measurement (*p* = 0.026). For maternal insulin levels, there was a significant main effect of oGTT (*p <* 0.001), but no significant interaction between group and oGTT (*p* = 0.070), indicating no relevant group differences (GDM vs. NGT) in maternal insulin levels.

At the follow-up time point, a repeated measures ANOVA on maternal blood glucose still showed an interaction of group with oGTT measurement (*p* = 0.041), with higher blood glucose levels in the GDM group. Similarly, for maternal insulin levels, it still showed an effect of oGTT (*p <* 0.001), and no significant interaction between group and oGTT (*p* = 0.126), indicating no relevant group differences (GDM vs. NGT) in the course of maternal insulin levels. Post hoc comparisons between the groups for all measurements are reported in Table 2.

### 3.3. Fetal Postprandial Brain Measurements

For fetal response latencies over the course of the oGTT (fasting, 1 h) in both groups, see Table 2 and Figure 2.

In the NGT group, there was no change in latency from fasting to 1 h during the oGTT at baseline (−13 ms, *p* = 0.465), and there was also no difference 4 weeks later after the intervention (+4 ms, *p* = 1.000).

In the GDM group, the fetal response latency increased from fasting to the postprandial state before and after intervention; however, this did neither reach significance before intervention (+52 ms, *p* = 0.285) nor after intervention (+54 ms, *p* = 0.715). The change in latency during oGTT did not differ between measurements before and after intervention. (*p* = 1.0).

Comparing the fetal latencies of the NGT and the GDM group before intervention, there was no difference in fasting state between GDM and NGT group (−4 ms, *p* = 1.0). In the postprandial state, fetuses in the GDM group reacted slower than those in the NGT group; however, this did not reach significance (+61 ms, *p* = 0.486).

After intervention, fetuses in the GDM group showed a slightly slower response in the fasting state compared to the NGT group (+31 ms, *p* = 1.0). In the postprandial state, fetuses in the GDM group reacted more slowly than those in the NGT group; however, this did not reach significance (+81 ms, *p* = 0.114).

## 4. Discussion

In the present study, we examined intervention during pregnancy in women with GDM and its impact on fetal event-related brain activity. In our previous studies, we detected an impact of GDM on fetal brain activity with specific differences between postprandial brain responses between fetuses of healthy mothers compared to fetuses of mothers with GDM [9,10]. In the present study, we aimed to investigate if these differences are reversible by dietary lifestyle intervention during pregnancy. We hypothesized that a lifestyle intervention with the goal of normalizing postprandial blood glucose levels below 7.8 mmol/L results in a shorter latency of the event-related brain response, which is a faster brain reaction time to the tone in fetuses of mothers with GDM.

We could not show that fetal brain responses were faster after 4 weeks of intensive dietary counselling and maternal blood glucose monitoring in fetuses of mothers with GDM. In fact, there was no change at all before and after intervention in the postprandial prolongation of latency in the fetuses of mothers with GDM. Taking into account the observed fetal latency difference in the GDM group before and after intervention of 2 ± 20 ms, the calculated sample size needed for showing that fetal latency improves during lifestyle intervention would be 787 fetuses of women with GDM.

A possible reason for this might be the timing of the intervention. Interventions starting after a GDM diagnosis, commonly after 24–28 weeks of gestation, might be clinically ineffective with regard to potential effects on brain insulin sensitivity, since the fetal brain is already programmed by the maternal metabolism. Epigenetic changes caused by altered maternal metabolism pursuant to Barkers hypothesis of fetal programming might start well earlier than gestational week 24 when GDM is diagnosed according to current standard of care [15]. Recent studies show that improving metabolic status of both parents before pregnancy has a positive impact on the health of mother and offspring [16]. Present knowledge about effects of preconceptional interventions on pregnancy outcome is still fragmentary but there are some trials in progress [17].

Our study has limitations. First, we were only able to analyze data of 8 women. Because of the long paradigm, repeated oGTT and advanced pregnancy at follow-up meant data collection was marked with dropouts. In addition, the specific study protocol may have introduced a bias in the included participants by selecting highly interested participants. Planning our study, we shortened the oGTT after intervention to simplify the study protocol, but still a number of participants had to be excluded because of artefacts in fMEG measurements or missing blood parameters. Due to the group sizes, we could only compare covariates related to maternal age and metabolism but were unable to include them in the main statistical analyses. The pre-pregnancy BMI of included GDM women ranged from 22 to 32, with a mean of 28. More severely overweight or obese women with GDM might have benefited even more from the intervention. Since women with an older maternal age and pregestational overweight/obesity are generally more at risk of GDM [18], including these in further research is of particular interest.

Maternal blood glucose levels in GDM women did not significantly change after the 4 weeks of intervention. Given the fact that insulin resistance increases during the second and third trimenon, which leads to higher postprandial blood glucose levels during the second and third trimenon, this still indicates a success of lifestyle intervention, as glucose levels did not deteriorate in the women with GDM. In addition to this non-deterioration, there are three variables that lead us to believe the intervention was successful, even though the group comparisons did not reach significance with our sample size—women with GDM showed less weight gain between visits than women with NGT, and HbA1c increased less strongly in women with GDM than in women with NGT (see Table 1). Moreover, there were no cases of neonatal macrosomia in the GDM group. Therefore, we assume the lifestyle intervention was effective. A control group of women with GDM without intervention would have further supported this conclusion, but was not included in our design for obvious ethical reasons.

Even though the lifestyle intervention seems to have been effective, the prolonged latencies of fetuses of mothers with GDM were not changed after intervention. We cannot fully exclude the possibility that an intervention that stabilizes and lowers the blood glucose levels of GDM women might have an effect on the fetal latencies also late in pregnancy, but we are not sure this is possible or even indicated at this point in pregnancy.

In this vein, while lifestyle interventions during pregnancy remain the primary tool in GDM therapy—decreasing the likelihood of postnatal depression, improving postpartum weight goals, and decreasing the risk of neonatal adiposity [7]—the current data do not support the efficacy of a dietary lifestyle intervention during the third trimester of pregnancy to reverse postprandial fetal brain programming induced by maternal gestational diabetes and insulin resistance.

Meta-analyses have indicated that more physical activity before or in early pregnancy [19] and diet or physical activity interventions before the 15th gestational week decreases GDM risk [20]. The DALI (vitamin D and lifestyle intervention for GDM prevention) lifestyle study found that a combined intervention before 20 weeks of gestation, focusing on motivational interviewing regarding physical activity and healthy eating in women with a BMI above 29, limited maternal gestational weight gain but did not reduce fasting glycemia [21]. Data from the Nurses Health Study indicate a reduction of GDM risk in relation to a healthy pregestational lifestyle [22]. We speculate that an earlier lifestyle intervention, ideally pre-pregnancy, as for example detailed by Barker and colleagues [17], is needed to reduce obesity and gestational diabetes risk associated with metabolically malprogrammed fetal brain in women of childbearing age.

## Figures and Tables

**Figure 1 nutrients-13-00556-f001:**
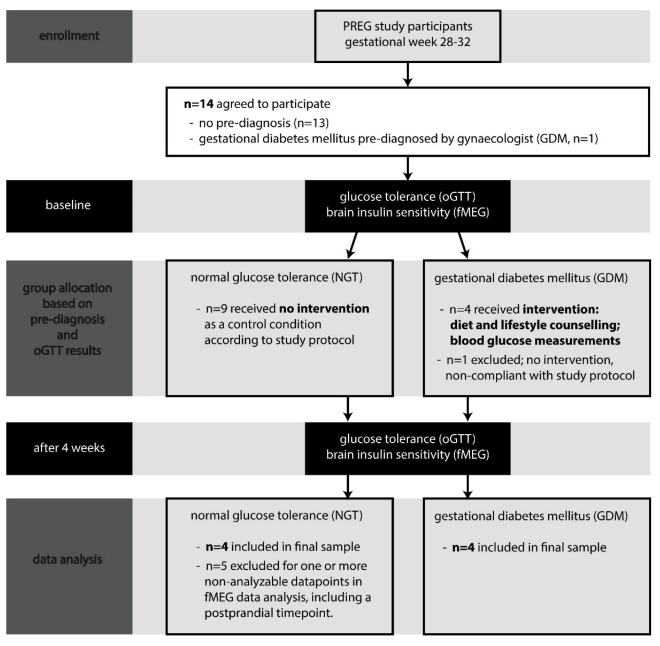
Flow diagram of the course of the study.

**Figure 2 nutrients-13-00556-f002:**
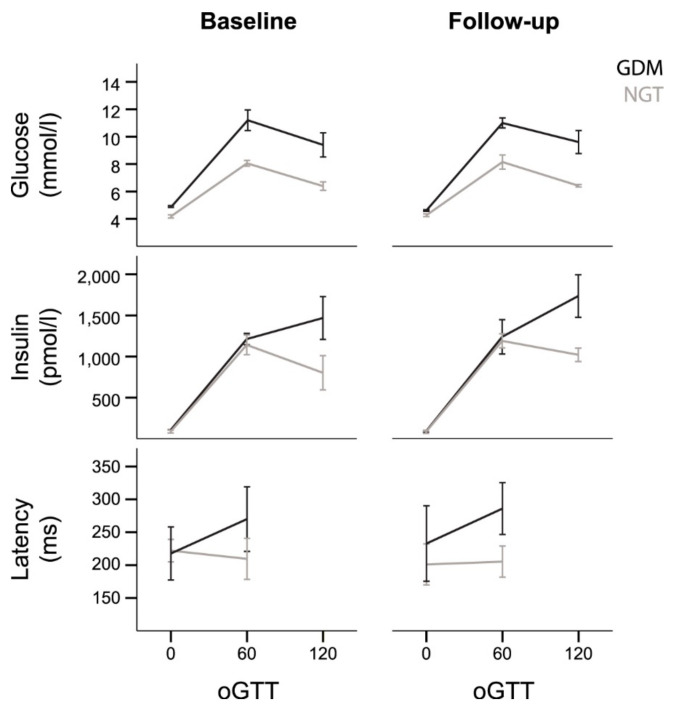
Maternal blood glucose levels, maternal blood insulin levels, and fetal auditory event-related latencies during the oral glucose tolerance test (oGTT). Grey lines: mothers with normal glucose tolerance (NGT); black lines: mothers with gestational diabetes mellitus (GDM). Error bars ±1 standard error.

**Table 1 nutrients-13-00556-t001:** Characteristics of study participants. Data are given as mean ± standard deviation.

Characteristic	NGT	GDM	*p*-Value (*)
Maternal age (years)	32.3 ± 3.0	37.0 ± 3.7	0.11
Gestational age (weeks) **	28.5 ± 1.0	28.8 ± 1.3	0.35
Pregestational BMI (kg/m^2^)	24.9 ± 2.3	28 ± 5	0.39
BMI (kg/m^2^) **	28 ± 1.8	30.4 ± 4.3	0.39
Relative weight gain (kg/week) **	8.6 ± 2.0	6.4 ± 2.2	0.20
Weight gain between time points (kg/m^2^)	1.25 ± 0.77	0.53 ± 0.85	0.20
HPA Index **	7.5 ± 1.5	7.76 ± 0.47	0.20
Parity (nulli-/multiparous)	3/1	1/3	0.15
C reactive protein (mg/dL)	0.47 ± 0.38	0.69 ± 0.4	0.38
HbA1c **	5 ± 0.1	5.4 ±0.1	0.03
HbA1c increase between time points	0.23 ± 0.19	0.05 ± 0.06	0.20
Neonatal birth weight (g)	3350 ± 272	3073 ± 584	0.39

* Wilcoxon–Mann–Whitney test was performed, as sample size was small; for parity, a chi squared test was performed. ** collected at baseline measurement timepoint. NGT: mothers with normal glucose tolerance, GDM: mothers with gestational diabetes, BMI: body mass index, HPA: habitual physical activity.

**Table 2 nutrients-13-00556-t002:** Metabolic characteristics of study participants. Data are given as mean ± standard deviation.

Category	Time Point	oGTT	NGT	GDM	*p*-Value (*)
Maternal glucose	Baseline	fasting	4.2 ± 0.2	4.9 ± 0.1	0.029
(mmol/L)		1 h	8.1 ± 0.4	11.2 ± 1.5	0.029
		2 h	6.4 ± 0.6	9.4 ± 1.8	0.029
	After 4 weeks	fasting	4.3 ± 0.2	4.6 ± 0.1	0.029
		1 h	8.1 ± 1.0	11.0 ± 0.7	0.029
		2 h	6.4 ± 0.1	9.6 ± 1.7	0.133
Maternal insulin	Baseline	fasting	86 ± 32	105 ± 6	0.686
(pmol/L)		1 h	1142 ± 240	1212 ± 137	0.486
		2 h	802 ± 416	1469 ± 519	0.343
	After 4 weeks	fasting	84 ± 37	91 ± 26	0.686
		1 h	1190 ± 172	1240 ± 418	0.686
		2 h	1021 ± 114	1736 ± 517	0.267
Fetal event-related	Baseline	fasting	222 ± 34	218 ± 81	1.000
response latency (ms)		1 h	209 ± 62	270 ± 98	0.486
	After 4 weeks	fasting	201 ± 62	232 ± 114	1.000
		1 h	205 ± 47	286 ± 79	0.114

* Mann–Whitney tests. oGTT: oral glucose tolerance test, NGT: mothers with normal glucose tolerance, GDM: mothers with gestational diabetes.

## Data Availability

The data presented in this study are available on request from the corresponding author. The data are not publicly available due to ethical regulations.

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
