# Peer review of "No Effect of Lifestyle Intervention during Third Trimester on Brain Programming in Fetuses of Mothers with Gestational Diabetes"

_nutrients, 2021, doi:10.3390/nu13020556_

Round 1

Reviewer 1 Report

The authors did not respond to the following question:

Could the level of amniotic fluid, placenta location or maternal obesity affect the fMEG recordings? Please explain and if yes please provide the appropriate data of your patients.

Reviewer 2 Report

The article has improved. The topic is important for the health of the mother and the newborn. The authors have addressed most of the recommendations and suggestions. However, the study limitations do not reflect all the weaknesses of the study, for example, no mention is made of possible biases. In data analysis, a multivariate analysis is not done. Also bibliographic references are very old, more than 50% of them are older than 10 years. There are only 7 references with an age of less than 5 years. About what the article is about, there is a lot of recent research that should be included.

Author Response

This manuscript is a resubmission of an earlier submission. The following is a list of the peer review reports and author responses from that submission.

Round 1

Reviewer 1 Report

Schleger et al. conducted a scientifically sound, yet small study about the influence of GDM on brain development. 

The authors pointed out themselves, that the limited number of participants is the biggest limitation of the study. The limited sample size, and most probably ethical considerations, also caused that one important control is missing: The GDM group without intervention. This group would have been important, especially with the small sample size, to prove that the intervention was effective. Although I agree with the authors, that a non-deterioration might be an indicator that there was an effect. It should be discussed that it is not impossible that an intervention which lowers the blood glucose and insulin resistance to the levels of healthy women could alter the fetal event-related response latency also late in pregnancy.

Table 1: In the anthropometric data of the participants, the ethnicity is missing.

Table 2: I don't understand why only in some of the data the Wilcoxon-Mann-Whitney-Test and in the others the t-test has been used. Can a normal distribution be assumed in a sample size of 4?

Reviewer 2 Report

The topic is interesting. A prevalent problem with significant health consequences is addressed.

In abstract, the conclusion is general and confusing. In the main text, the conclusion makes recommendations that are not based on the results. This should move to the discussion section

Most of the bibliographic references (60%) are older than 5 years and many of them are even older than 10 years

The inclusion and exclusion criteria used for the selection of subjects are not reported.

It must be specified what the intervention specifically consisted of. More information about the intervention is required.

The authors mention some limitations associated with the small number of participating women, but there are other limitations. For example, the differences that may exist between participants in the control group and the intervention group are not reported. Variables or factorspersonal history, parity, etc. () that can influence the results and can be confusing are not taken into account

The characteristics of the samples (neither the control group nor the intervention group) are not reported.

Reviewer 3 Report

In the manuscript authors analyzed the effect of GDM treatment (diet+physical activity) on the fetal brain activity durig OGTT. The study might be of interest to readers, however, in my opinion very small group of participants in each arm (4) do not allow to make final conclusions.

  1. The study definitely need to be performed in a larger sample size. Maybe not on 787 fetuses but 30-40 participants.
  2. What was the regimen of daily physical activity among patients with GDM?
  3. Why the second fMEG data analysis was performed only after 4 weeks? The mean GA was 28 weeks so it is plausible to perform the examination at 32 and 36 gestational weeks.
  4. Could the level of amniotic fluid, placenta location or maternal obesity affect the fMEG recordings? Please explain and if yes please provide the appropriate data of your patients.
